# Temporal Trends and Differences in Sexuality among Depressed and Non-Depressed Adults in the United States

**DOI:** 10.3390/ijerph192114010

**Published:** 2022-10-27

**Authors:** Weiya Li, Yu Wang, Mingyu Xu, Yingxue Liao, Haofeng Zhou, Huan Ma, Qingshan Geng

**Affiliations:** 1Guangdong Cardiovascular Institute, Guangdong Provincial People’s Hospital, Guangdong Academy of Medical Sciences, Guangzhou 510317, China; 2School of Medicine, South China University of Technology, Guangzhou 510641, China

**Keywords:** sexuality, age at first sexual intercourse, sexual frequency, sexual orientation, depression

## Abstract

This study aimed to examine temporal trends and differences in sexuality between depressed and non-depressed adults aged 18–59 in the United States from 2005 to 2016. A total of 21,437 people (5432 with depression) were enrolled in this cross-sectional study. From 2005–2008 to 2013–2016, the average age at first sexual intercourse decreased, while the proportion of normal frequency of sexual activity and heterosexual sexual orientation increased among all the participants. Some differences in sexuality were found between the depressed and non-depressed groups. The average age at first sexual intercourse (*p* < 0.001), the proportion of normal frequency of sexual activity (*p* < 0.001), and heterosexual sexual orientation (*p* < 0.001) were lower in depressed participants, and the differences did not change over time (*p*_for trend_ = 0.926 of average age at first sexual intercourse, *p*_for trend_ = 0.823 of normal frequency of sexual activity, *p*_for trend_ = 0.926 of heterosexual sexual orientation). Moreover, these differences were associated with marital status (*p*_for interaction_ < 0.001 by average age at first sexual intercourse), employment status (*p*_for interaction_ < 0.001 by average age at first sexual intercourse), education status (*p*_for interaction_ = 0.023 by heterosexual sexual orientation) and family income status (*p*_for interaction_ = 0.013 by average age at first sexual intercourse and *p*_for interaction_ = 0.017 by normal frequency of sexual activity). In conclusion, the study found that the age at first sexual intercourse decreased and the frequency of sexual intercourse increased in all the participants, and differences in sexuality between depressed and non-depressed participants were present; however, these differences had no further increase or decrease during the 12-year period. These differences were associated with marital status, employment status, education status, and family income status. These findings show differences in sexuality between depressed and non-depressed patients but are somewhat different from previous studies; the results may provide directions for future research and social work.

## 1. Introduction

Sexual health is a state of physical, emotional, mental, and social well-being in relation to sexuality [1]. Many previous studies have verified that sexuality is vital to proper and healthy human development [2]. Sexuality has been proven to be related to unwanted pregnancies, physical health [1,3], risk of fatal coronary events [4], cancers [5], sexually transmitted diseases (STDS) [6,7,8,9], mental health, and quality of life [10]. Sexual minorities (homoerotism, bisexual) carry a greater risk of suffering from chronic disease, suicide attempts, and all-cause mortality than heterosexuals [11,12,13]. Similarly, a younger age at first sexual intercourse can also cause more unintended pregnancies and HPV transmission and higher prevalence ratios for precancerous lesions among women [14,15,16].

Sexuality is influenced by the interaction of biological, psychological, social, economic, political, cultural, legal, historical, religious, and spiritual factors. Many studies have proven that depression is associated with sexual activity, sexual orientation, and age at first sexual intercourse [17,18,19]. A high rate of depressive disorder and depressive symptoms was demonstrated in minority youth and adolescents with experience of sexual intercourse [19,20]. Depression could cause severe sexual dysfunction, and sexuality was also believed to be a key factor for depression [21,22,23]. Given the strong association between sexuality and depression, as well as the increasing prevalence of depression in recent years, describing the temporal trends of sexuality between depressed and non-depressed individuals becomes particularly important for public health. However, there are still no studies showing the trends and differences in the related indicators of sexual frequency, sexual orientation, and time of first sexual intercourse between depressed and non-depressed people. We aimed to explore these differences by using data from the National Health and Nutrition Examination Survey (NHANES) and to assess whether these differences are related to age, race/ethnicity, marital status, employment status, education status, and family income status.

## 2. Methods

### 2.1. Study Design and Study Population

Data from the National Health and Nutrition Examination Survey (NHANES) were used for this research. The NHANES collects participant-reported data from a nationally representative sample of US residents. The complete protocols and methods have been previously reported. In this analysis, data from people aged between 18 and 59 were extracted from the three 4-year NHANES cycles conducted between 2005 and 2016. The Patient Health Questionnaire 9 (PHQ-9) scale was used to define depression in our study. The respondents were asked 9 questions about specific symptoms, assigning values of 0 to 3 points (0—not at all, 1—several days, 2—more than half of the days, 3—nearly every day), with a higher score on each item representing more frequently being affected by the symptom [24]. PHQ-9 has been demonstrated to be a reliable predictor of depression and a score ≥ 5 indicates the presence of depression [24]. Any missing answers to the 9 questions were considered incomplete data and were excluded in our study. All the participants with a PHQ-9 score greater than 5 were defined as depressed, while others were defined as non-depressed.

### 2.2. Definition of Sexuality and Other Variables

We assessed sexuality in terms of age at first sexual intercourse, frequency of sexual activity, and sexual orientation. Age at first sexual intercourse was assessed by asking the participants how old they were when they first had sex. Frequency of sexual activity was assessed by asking how many times they had sex per year. Having sex between 52 and 365 times a year was defined as normal, while having sex less than 52 times or more than 365 times a year was defined as abnormal. Sexual orientation was assessed by asking how participants would describe their sexual orientation. In any of these questions, participates who refused to answer the question or answered “don’t know” were considered as missing data.

Information on age, race/ethnicity (including Non-Hispanic White, Non-Hispanic Black, Hispanic, and Others), and marital status (including married, living with partner, never married, widowed, divorced or separated) was collected. Married or living with a partner were defined as living together, while being never married, widowed, divorced, or separated was defined as living alone). Employment status (including employed and unemployed) and education status (including below high school, high school graduate or general educational development, and some college or above) were extracted from the NHANES database. The Poverty Impact Ratio (PIR) was used to estimate family income status (<1.3 was considered comfortable, while ≥3.5 was considered poor) [25].

### 2.3. Statistical Analyses

Means were estimated for risk factors measured on a continuous scale, and the proportion was estimated for categorical variables.

Differences in sexuality between the depressed and non-depressed participants were computed by using a multi-variable logistic regression model. To examine the temporal trends for sexuality, *p* values for differences between the depressed and non-depressed participants across calendar periods were derived by adding an interaction term between depression and calendar period to the model. For each type of sexuality, respondents with missing data were excluded from the analyses.

Subgroup analyses were conducted by age group (18–38, 39–59 years old), race/ethnicity, marital status, employment status, education status, and family income status. To assess whether sexuality differences between the depressed and non-depressed participants in temporal trends differed from the subgroups above, we added three-way interaction terms (depression, calendar period, and subgroup variables) to the model.

To obtain nationally representative values, all analyses were weighted using the NHANES sample weights, thus taking account of the complex sampling design. Analyses were performed in R version 4.1.2.

## 3. Results

### 3.1. Baseline Characteristics

Table 1 shows the baseline characteristics of 21,437 (5432 with depression) participants using data from the NHANES 2005–2016. Similar distributions existed in age group, race/ethnicity, marital status, employment status, education status, and family income status between depressed and non-depressed individuals. Compared with the non-depressed group, there were more women (59.94% vs. 47.93%) in the depressed group. The proportion of PIR < 1.30 in the depressed group was higher than that in the non-depressed group (31.61% vs. 18.31%), while the proportion of PIR ≥ 3.50 was lower than that in the non-depressed group (27.98% vs. 44.96%). Results from the 2005–2008, 2009–2012, and 2013–2016 periods are provided in Appendix A.

### 3.2. Trends of Sexuality from 2005–2008 to 2013–2016

From 2005–2008 to 2013–2016, the average age at first sexual intercourse decreased from 16.65 to 16.46 for depressed participants and from 17.59 to 17.43 for non-depressed participants. The proportion of normal frequency of sexual activity increased from 24.13% to 25.47% in depressed participants and 28.09% to 29.74% in non-depressed participants. The proportion of heterosexual sexual orientation in both groups also changed, with an increasing proportion from 62.88% to 68.48% in depressed participants and from 70.89% to 77.49% in non-depressed participants (Figure 1).

### 3.3. Differences in Sexuality between Depressed and Non-Depressed Participants

Significant differences existed in sexuality between depressed and non-depressed participants. The average age at first sexual intercourse in the depressed group was nearly 1 year lower than that in the non-depressed group (16.50 vs. 17.46, *p* < 0.001), and the proportions of normal frequency of sexual activity and heterosexual sexual orientation were 5.3% (24.21% vs. 29.44%, *p* < 0.001) and 9.42% (66.10% vs. 75.51%, *p* < 0.001) lower than those in the non-depressed group, respectively (Table 2). Although the three indicators of sexuality changed, the differences between depressed and non-depressed participants did not increase or decrease over time (*p*_for trend_ = 0.823 of average age at first sexual intercourse, *p*_for trend_ = 0.926 of normal frequency of sexual activity, *p*_for trend_ = 0.926 of heterosexual sexual orientation; Table 3).

### 3.4. Subgroup Analysis of Differences in Sexuality

The difference in the average age at first sexual intercourse was found to be associated with marital status (*p*_for interaction_ < 0.001), employment status (*p*_for interaction_ < 0.001), and family income status (*p*_for interaction_ = 0.013). Regardless of marital status, employment status, or family income status, depressed participants had a lower average age at first sexual intercourse than non-depressed participants, and the differences were statistically significant (Table 3, Figure 2A–C).

The difference in the normal frequency of sexual activity was of marginal significance by education status (*p*_for interaction_ = 0.062) and was found to be influenced by family income status (*p*_for interaction_ = 0.017, Table 3). Among participants who had received a high school or general educational development diploma (*p* = 0.009) or had moderate to high PIR (*p* = 0.019 for PIR ≤ 1.3 and *p* = 0.009 for 1.3 < PIR < 3.5), depressed participants were more likely to have a normal frequency of sexual activity than non-depressed participants, while non-depressed participants were more likely to have a normal frequency of sexual activity among those who had received a college degree (*p* = 0.033) or had a low PIR (*p* = 0.036) (Table 3, Figure 2D–E). The proportion of normal frequency of sexual activity was also higher among depressed participants with an education status of below high school; the difference nearly reached a statistically significant level (*p* = 0.053, Table 3, Figure 2D). Moreover, among participants with a PIR ≥ 3.5, the depressed participants were less likely to have a normal frequency of sexual activity than the non-depressed participants; the difference nearly reached statistical significance (*p* = 0.036, Table 3, Figure 2E).

The difference in the proportion of heterosexual sexual orientation was also found to be influenced by education status (*p*_for interaction_ = 0.023, Table 3). Among participants with a college degree or above, the proportion of heterosexual orientation was lower among those with depression, while the opposite phenomenon was found in those with lower educational levels (Table 3, Figure 2). This means that participants with high levels of education represented a higher proportion of sexual minorities in the depressed than in the non-depressed group, while participants with only low–moderate education levels represented a lower proportion of sexual minorities in the depressed group.

## 4. Discussion

Our study shows trends and differences in sexuality between depressed and non-depressed adults aged 18–59 years in the United States from 2005 to 2016. The average age at first sexual intercourse decreased in both groups, while the proportion of normal frequency of sexual activity and the proportion of heterosexual orientation increased. Compared with non-depressed participants, depressed participants were younger at their first sexual intercourse and had a lower proportion of normal frequency of sexual activity and heterosexual orientation. However, differences in sexuality between the two groups did not change over time. In addition, our study found that the differences in sexuality between the two groups were associated with marital status, education status, employment status, and family income status.

Previous researchers have shown that having sex for the first time between the ages of 16 and 24 is associated with better physical and mental health and a lower risk of depression [19,26]. In our study, the average age at first sexual intercourse was 16.5 years in depressed patients and 17.5 years in non-depressed participants, which is consistent with previous studies. The results also suggest that delaying the age at first intercourse may be beneficial to health, similar to the findings of previous studies [27]. According to our results, the average age at first sexual intercourse decreased between 2005 and 2013, which was matched by an increase in the proportion of participants in the depression group and in part reflected the potential association between early sexual intercourse and depression. Engaging in sexual activity at a younger age may indicate the presence of less comprehensive sexual education programs and can increase the incidence of STDs, especially in the United States compared to other western nations [28,29]. In addition, women are more likely to be subjected to coercion, abuse, and even unintended pregnancy and abortion in their early years, and, due to physiological differences, they are more sensitive to emotional stimuli than men, which leads to a greater prevalence of depression [30,31].

Marital status, education status, employment status, and family income status have long been reported to be related to depression [32,33,34], but few studies have focused on the relationship between these factors and age at first sexual intercourse. Previous studies have found that young people in the United States are increasingly delaying their first sexual activity [35]. This seems to be at odds with our findings, but it was consistent that depressed people experienced their first sexual activity at an earlier age. These results may suggest that the effect of depression on sexuality may work in both directions: it is possible to engage in sexual activity too early and become depressed, or to delay sexual activity because of depression. Further prospective studies are needed to explore the exact link.

Inconsistently with previous studies, the frequency of sexual activity increased from 2005 to 2016 [36]. This may be due to differences in the timing of the survey and the frequency at which normal sex was defined. It is not possible to generalize the finding that the frequency of sexual activity increased. Our results only showed a slight increase in the proportion of people having sex more than once per week, and we did not calculate the frequency of sexual activity by age. Our results also show that depressed participants were less likely to report having sex at least once a week than non-depressed participants. Sexual activity can lower the heart rate and blood pressure [4,37] while also reducing stress by promoting oxytocin release [38]; less sex is associated with increased mortality and self-reported poor health [39,40], so maintaining a certain frequency of sexual activity is beneficial to improve mood and promote psychosomatic health. Previous studies showed that the average frequency of sexual activity for American 20-year-olds is 80 times per year, while that for 60-year-olds is 20 times per year [36]. Based on this, it can be inferred that the normal frequency of sexual activity for middle-aged American adults (around 40 years old) is 50 times per year. Therefore, considering that the average age of the participants was around 39, it is reasonable that we defined a normal frequency of sexual activity as at least once a week, and the results are credible. Moreover, a previous study and clinical practice also found that an active sex life is detrimental to health, and frequent sex may indicate potential psychological problems and increase the risk of organ damage, cardiovascular disease, or depression [41]. Therefore, it is possible to classify participants who have sex more than 365 times per year as abnormal. However, the results of this study can only show that there is a difference in sexual frequency between depressed and non-depressed participants, and it cannot be concluded that low sexual frequency causes depression or that depression reduces sexual frequency. It is necessary to use a large sample size and a prospective study to further elucidate the causal relationship between depression and the frequency of sexual activity and to explore the optimal frequency of sexual activity.

Previous studies have shown that those with a lower economic status (including no job, low income, low education level) are more likely to engage in unhealthy sexual behavior and have a lower frequency of sexual activity [42,43], which is not exactly consistent with our findings. Our results also suggested that the proportion of normal sexual activity was lower in the low-education subgroup, whereas only non-depressed participants showed a reduction in the proportion of normal sexual activity in the low-income subgroup analysis, and the opposite was true for depressed participants. This phenomenon reflects the unclear cause-and-effect relationship between depression, sexual activity frequency, and income statues; however, it could also reflect the notion that the effect of depression on sexuality among people with different education levels and PIR is inconsistent.

Sexual orientation, especially sexual minorities, is also strongly associated with depression [17,44]. Considering that there may be various reasons for participants to hide their sexual orientation when completing the survey, we calculated the proportion of participants who reported their orientation as heterosexual, which indirectly reflected the proportion of minorities. Our results also showed that the depressed group included a lower proportion of heterosexual individuals, which means a higher proportion of sexual minorities. Sexual minorities have a higher incidence of STDs, which may increase the risk of depression [45]. In addition, depression caused by poor relationships with the opposite sex may also lead to a change in sexual orientation. An interesting observation of our study was that, among highly educated participants, the proportion of heterosexual sexual orientation was lower in the depressed group than in the non-depressed group. Participants with higher education degrees may have faced more stress and had more open sexual attitudes, which could explain the results. Although the causal relationship between sexual orientation, depression, and education level is also not clear, this conclusion suggests that sexual orientation and depression among students at different levels of school is a social issue worthy of attention.

### 4.1. Strength

The strengths of this study lie in the comprehensive nature of the analysis. It provides detailed, reliable, and nationally representative temporal findings on the differences in sexuality among depressed and non-depressed individuals in the US. The NHANES surveys are large and subject to rigorous quality control, and, as such, they represent a high-quality data source. Our results demonstrate differences in sexuality between depressed and non-depressed individuals, helping to identify potential depression-related factors and providing guidance for future research.

### 4.2. Limitations

This study also has some limitations. First, the NHANES was a cross-sectional study, and some data are not available, such as the order in which depression and sexual activity occurred. In addition, some other symptoms could also influence sexuality, and there might also be an interaction occurring between age, race/ethnicity, marital status, employment status, education status, and family income status that we have not taken into account; therefore, our results cannot accurately show the causal relationship between these factors, sexuality, and depression, but can only reflect some correlation. Secondly, each survey included data from a different sample of participants, so sampling error could affect the comparisons over time. We weighted our results according to recommendations, so as to minimize the sampling error. Thirdly, some data from the NHANES are self-reported, leading to a higher potential for misreporting than in raw clinical data.

## 5. Conclusions

From 2005–2008 to 2013–2016, the average age at first sexual intercourse decreased while the proportion of normal frequency of sexual activity and heterosexual sexual orientation increased among both depressed and non-depressed participants. Differences in sexuality were present between depressed and non-depressed individuals, but the differences did not change over time. The average age at first sexual intercourse and the proportions of normal frequency of sexual activity and heterosexual sexual orientation were lower in depressed participants, and the differences between depressed and non-depressed participants were associated with marital status, employment status, education status, and family income status. Our study provides detailed and nationally representative findings on the differences in sexuality between depressed and non-depressed adults in the United States. A prospective study with a large sample size should be conducted to examine the associations between socioeconomic factors, depression, and sexuality in the future, and to further explore the optimal sexuality model.

## Figures and Tables

**Figure 1 ijerph-19-14010-f001:**
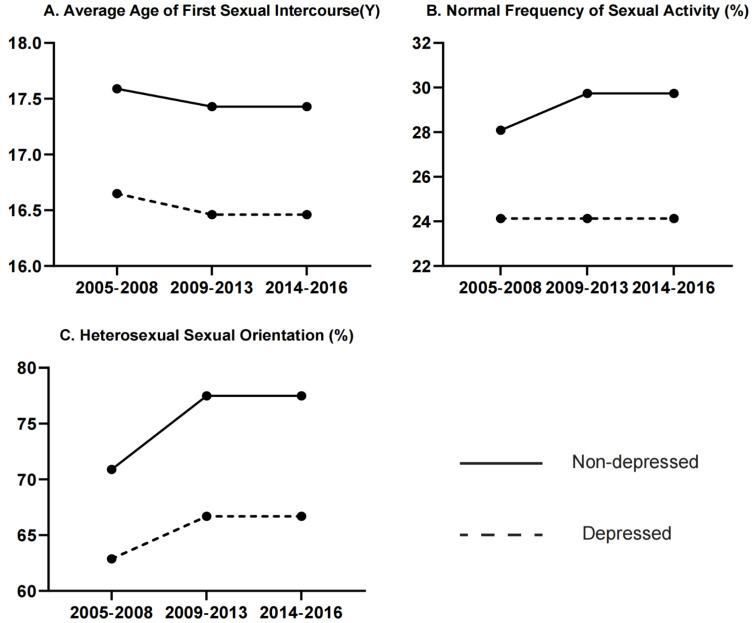
Trends in sexuality of depressed and non-depressed participants from the NHANES surveys over time. (**A**): Average age at first sexual intercourse (Y); (**B**): normal frequency of sexual activity (%); (**C**): heterosexual sexual orientation (%). Dotted line indicates “depressed” and solid line indicates “non-depressed” participants.

**Figure 2 ijerph-19-14010-f002:**
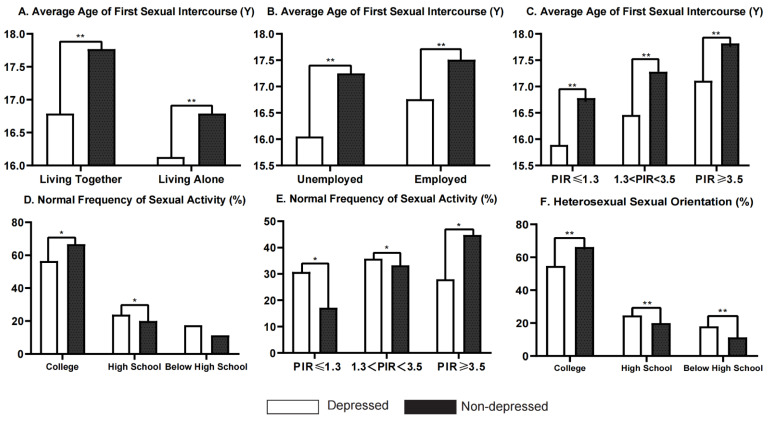
Subgroup analysis of differences in sexuality between depressed and non-depressed participants from NHANES 2005–2016. (**A**): Differences in average age at first sexual intercourse by marital status; (**B**): differences in average age at first sexual intercourse by employment status; (**C**): differences in average age at first sexual intercourse by family income status; (**D**): differences in normal frequency of sexual activity by education status; (**E**): differences in normal frequency of sexual activity by family income status; (**F**): differences in heterosexual sexual orientation by education status; hollow box indicates “depressed” and solid box indicates “non-depressed” participants; college means “some college or above”; high school means ”high school graduate or general educational development”; ** *p* ≤ 0.001, * *p* < 0.05.

**Table 1 ijerph-19-14010-t001:** Baseline characteristics of depressed and non-depressed participants.

	Depressed (*n* = 5432)	Non-Depressed (*n* = 16,005)
**Age (Y)**	39.05 (38.60–39.50)	38.44 (38.07–38.81)
**Female (%)**	59.94	47.94
**Age Group (%)**		
18–38 (Y)	46.63 (44.72–48.53)	49.73 (48.25–51.21)
39–59 (Y)	53.37 (51.47–55.28)	50.27 (48.79–51.75)
**Race/Ethnicity (%)**		
Non-Hispanic White	63.01 (59.61–66.41)	65.18 (62.15–68.21)
Non-Hispanic Black	13.36 (11.34–15.38)	11.58 (10.00–13.16)
Hispanic	16.43 (14.20–18.67)	15.75 (13.59–17.91)
Others	7.19 (6.17–8.22)	7.49 (6.65–8.33)
**Marital Status (%)**		
Living together	51.97 (49.99–53.95)	63.78 (62.29–65.27)
Living alone	44.14 (42.15–46.14)	32.52 (31.13–33.91)
**Employment Status (%)**		
Employed	60.74 (58.57–62.92)	79.21 (78.11–80.32)
Unemployed	39.25 (37.07–41.42)	20.74 (19.63–21.84)
**Education Status (%)**		
Some college or above	52.00 (49.48–54.53)	62.51 (60.57–64.45)
High school graduate or GED	23.84 (21.97–25.71)	19.92 (18.85–21.00)
Below high school	19.68 (18.03–21.33)	13.07 (11.80–14.34)
**Family Income Status (%)**		
PIR < 1.30	31.61 (29.39–33.83)	18.31 (16.80–19.82)
PIR: 1.30–3.50	34.58 (32.52–36.64)	31.33 (29.94–32.72)
PIR ≥ 3.50	27.98 (25.81–30.16)	44.96 (42.74–47.18)

GED: general educational development; PIR, Poverty Impact Ratio. Values are means for continuous variables and percentages for categorical variables; values in brackets represent the confidence interval of the corresponding variables.

**Table 2 ijerph-19-14010-t002:** Differences in sexuality between depressed and non-depressed participants.

	Depressed	Non-Depressed	Depressed vs. Non-Depressed	*p*
**Average Age at First Sexual Intercourse (Y)**	16.50 (16.34–16.66)	17.46 (17.34–17.59)	−0.96 (−1.14 to −0.79)	<0.001
**Normal Frequency of Sexual Activity (%)**	24.21 (22.75–25.67)	29.44 (28.31–30.58)	−5.3 (−7.06 to −3.41)	<0.001
**Heterosexual Sexual Orientation (%)**	66.10 (64.22–67.98)	75.51 (74.51–76.52)	−9.42 (−11.52 to −7.31)	<0.001

Values are means for continuous variables and percentages for categorical variables; values in brackets represent the confidence interval of the corresponding variables. Depressed vs. non-depressed: difference in depressed minus non-depressed; values in brackets represent the confidence interval of difference between the depressed and non-depressed participants.

**Table 3 ijerph-19-14010-t003:** Temporal trends and differences in sexuality between depressed and non-depressed individuals by subgroup.

	Average Age at First Sexual Intercourse (Y)		Normal Frequency of Sexual Activity (%)		Heterosexual Sexual Orientation (%)	
** *p* _for trend_ **	0.926		0.823		0.552	
**By Age**	** *p* **	** *p* _for interaction_ **	** *p* **	** *p* _for interaction_ **	** *p* **	** *p* _for interaction_ **
18–38	<0.001	0.818	0.136	0.136	<0.001	0.647
39–59	<0.001		0.002		0.003	
**By Race/Ethnicity**	** *p* **	** *p* _for interaction_ **	** *p* **	** *p* _for interaction_ **	** *p* **	** *p* _for interaction_ **
Non-Hispanic White	<0.001	0.316	0.004	0.323	<0.001	0.404
Non-Hispanic Black	0.003		0.229		<0.001	
Hispanic	0.001		0.001		0.002	
Others	<0.001		0.244		<0.001	
**By Marital Status**	** *p* **	** *p* _for interaction_ **	** *p* **	** *p* _for interaction_ **	** *p* **	** *p* _for interaction_ **
Living Together	<0.001	<0.001	0.023	0.725	<0.001	0.846
Living Alone	<0.001		0.772		<0.001	
**By Employment Status**	** *p* **	** *p* _for interaction_ **	** *p* **	** *p* _for interaction_ **	** *p* **	** *p* _for interaction_ **
Employed	<0.001	<0.001	0.163	0.389	<0.001	0.460
Unemployed	<0.001		0.003		<0.001	
**By Education Status**	** *p* **	** *p* _for interaction_ **	** *p* **	** *p* _for interaction_ **	** *p* **	** *p* _for interaction_ **
Some College or Above	<0.001	0.446	0.033	0.062	<0.001	0.023
High School	<0.001		0.009		<0.001	
Below High School	0.003		0.053		0.001	
**By Family Income Status**	** *p* **	** *p* _for interaction_ **	** *p* **	** *p* _for interaction_ **	** *p* **	** *p* _for interaction_ **
PIR ≤ 1.3	<0.001	0.013	0.019	0.017	<0.001	0.271
1.3 < PIR < 3.5	<0.001		0.009		<0.001	
PIR ≥ 3.5	<0.001		0.036		0.004	

GED: general educational development; PIR, Poverty Impact Ratio.

## Data Availability

Data are contained within the article.

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
