# Peer review of "Temporal Trends and Differences in Sexuality among Depressed and Non-Depressed Adults in the United States"

_ijerph, 2022, doi:10.3390/ijerph192114010_

Round 1
Reviewer 1 Report
This manuscript presents the differences in sexuality between depressed and non-depressed. The advantage of this study is the number of participants of more than 21,000 people who were studied over a 12-year period, which allowed the authors to show temporal trends in sexuality. The manuscript is interesting and generally well written, but there are some points that need to be clarified and completed prior to publication.
Detailed comments:
1. The introduction should be more related to the topic and objectives of this study, which are temporal trends in sexuality and the relationship between sexuality and depression. Please clearly indicate the problem and the rationale for this study.
2. In the Methods section, consider adding details about the factors that identified depression in participants.
3. Please provide the reference or justify why „Having sex more than 52 times a year was defined as having a normal sexual frequency”. Has the upper limit above which excess sexual frequency has occurred has been taken into account? Other studies have shown that hypersexuality has serious health, psychological and social consequences and can lead to depression.
4. In section 2.3. the sentence „Differences on sexuality between depressed and non-depressed were computed on the absolute scale using linear regression analyses” is not understandable. Linear regression can judge the relationship rather than the difference between the variables.
5. In table 1, please provide the standard deviation or the confidence interval for the mean age.
6. In the notes under tables 1 and 2, please explain what the values in brackets for continuous and categorical variables mean.
7. What statistical test was used to test the significance of the differences shown in Figure 2 and Table 2?
8. In the notes below table 3, please explain the abbreviations NH-W and NH-B.
9. If linear regression analysis was used, why were the regression coefficients and R2 not reported, which would indicate the direction and strength of changes over time?
10. Please consider rewriting the last sentence in the conclusions section. The wording "reliable" may be debatable. Please specify what directions this work provides for future research.
Author Response
Response to Reviewer 1 Comments
Thanks for your careful reviewing and insightful suggestions. We have carefully revised our manuscript according to the comments. Our point-by-point responses are provided below.
Detailed comments:
Point 1: The introduction should be more related to the topic and objectives of this study, which are temporal trends in sexuality and the relationship between sexuality and depression. Please clearly indicate the problem and the rationale for this study.
Response 1: Thank you very much for this important comment. We have revised the Introduction part according to your valuable comments, emphasizing the relationship between sexuality and depression, declaring the current problems and the values of our study. (Page: 2. As revised content marked up in the “Track Changes” mode).
Point 2: In the Methods section, consider adding details about the factors that identified depression in participants.
Response 2: Thanks for your valuable advice. We have added details about the factors that identified depression in participants in the Methods section. (See Page: 2 Methods part).
Point 3: Please provide the reference or justify why „Having sex more than 52 times a year was defined as having a normal sexual frequency”. Has the upper limit above which excess sexual frequency has occurred has been taken into account? Other studies have shown that hypersexuality has serious health, psychological and social consequences and can lead to depression.
Response 3: Thanks for this constructive advice and we completely agree with you. Just as the Reviewer mentioned hypersexuality also has serious health, psychological and social consequences and can lead to depression. We couldn't agree more with the Reviewer and we realized that our grouping model might be inappropriate. Thank you for your reminding! According to your suggestion, we re-defined all the participates with sex between 52 and 365 times a year as normal, while having sex less than 52 times or more than 365 times a year was defined as abnormal. All the analyses were reperformed. The specific statistical results showed some changes, but the research conclusion did not change significantly, which was also shown in the latest manuscript version.
In addition, the frequency of sexual activity provided by the NHANES database is not the specific number of times, but is grouped according to less than 1 time/year, 1-12 times/year, 12-51 times/year, 52-103 times/year 104-365 times/year or more. As mentioned in the Reference 36 (Twenge JM, Sherman, RA, Wells BE Declines in sexual frequency among American adults. 1989-2014. Arch Sex Behav. 2017; 46(8):2389-2401). Previous studies have shown that the frequency of sexual activity for young people (about 20 years old) is about 80 times per year, and that for the older adults (about 60 years old) is about 20 times per year. On the whole, the frequency of sexual activity decreases with the increase of age. Considering that the average age of our study group is about 40 years old, which is in the middle of 20-60 years old, it is estimated that the average sex frequency of this age group is about 50 times per year. So choosing more than 52 times per year as the normal sex frequency may not be accurate, but it is feasible.
Point 4: In section 2.3. the sentence „Differences on sexuality between depressed and non-depressed were computed on the absolute scale using linear regression analyses” is not understandable. Linear regression can judge the relationship rather than the difference between the variables.
Response 4: Thank you for this important comment, I am very sorry for the description error. The statistical method used here is logistic regression. It has been modified in the Methods section. (At the top of Page 3).
Point 5: In table 1, please provide the standard deviation or the confidence interval for the mean age.
Response 5: Thank you for your valuable advice. We revised it according to your comments and present it in the latest manuscript. (See Page: 11 Table1).
Point 6: In the notes under tables 1 and 2, please explain what the values in brackets for continuous and categorical variables mean.
Response 6: Thank you very much for your careful reviewing of our manuscript. We have added the interpretation what the values in brackets for continuous and categorical variables mean under tables 1 and 2.
Point 7: What statistical test was used to test the significance of the differences shown in Figure 2 and Table 2?
Response 7: Thank you for this critical comment. We used logistic regression calculate the significance of the differences shown in Figure 2 and Table 2.
Point 8: In the notes below table 3, please explain the abbreviations NH-W and NH-B.
Response 8: Thank you very much for your careful reviewing. We have replaced the abbreviations NH-W and NH-B as Non-Hispanic White and Non-Hispanic Black in the Table 3.
Point 9: If linear regression analysis was used, why were the regression coefficients and R2 not reported, which would indicate the direction and strength of changes over time?
Response 9: Thank you for this important comment, I am very sorry for the description error. The statistical method we used is logistic regression. It has been modified in the Methods section. (At the top of Page 3). P for trend is used to assess the direction of changes over time.
Point 10: Please consider rewriting the last sentence in the conclusions section. The wording "reliable" may be debatable. Please specify what directions this work provides for future research.
Response 10: We have revised the last sentence in the conclusions section and deleted the words "reliable" and specified the directions this work provided for future research.

Reviewer 2 Report
Thank you for your paper on a very interesting and important topic.
Please complete the introduction part with relevant theoretical background.
Please explain the criteria for the subgroups from 18-38 and 39-59.
Please explain the differences in age differences for first sexual intercourse and type of sexual orientation.
Author Response
Response to Reviewer 2 Comments
Thanks for your valuable comments and suggestions. We reply the comments point-by-point as following.
Detailed comments:
Point 1: Please complete the introduction part with relevant theoretical background.
Response 1: Thank you very much for this important comment. We have revised the Introduction part according to your valuable comments.
Point 2: Please explain the criteria for the subgroups from 18-38 and 39-59.
Response 2: Thank you for this important comment. The NHANES data on sexual characteristics, such as frequency of sexual activity and sexual orientation were only investigated in people aged 18-59 years, so our analysis was limited to this age group. Further statistical analysis revealed that the average age of the participants in our study was 38-39 years, so we divided the participants into two age groups for further analysis by using 38 as the dividing point.
Point 3: Please explain the differences in age differences for first sexual intercourse and type of sexual orientation.
Response 3: As shown in Table 3, our analysis showed that age group is not related with the differences of sexuality between the depressed and non-depressed participants. There have been few previous studies on this topic, and one possible reason for the differences in sexuality among the participants in different age groups is: The different social environments in which the participants were born and grew up. This point is mentioned in the discussion section. Due to space limitation, the results of this part are not presented more intuitively in the form of figures. This may also be the direction of further research in the future. Thank you very much for your question!
